# Fatal Association of Mirror and Eisenmenger Syndrome during the COVID-19 Pandemic

**DOI:** 10.3390/medicina57101031

**Published:** 2021-09-28

**Authors:** Viorica Radoi, Lucian Gheorghe Pop, Nicolae Bacalbasa, Anca Maria Panaitescu, Anca Marina Ciobanu, Dragos Cretoiu, Oana Daniela Toader

**Affiliations:** 1Department of Obstetrics and Gynecology, National Institute of Mother and Child Care Alessandrescu-Rusescu, 020395 Bucharest, Romania; viorica.radoi@yahoo.com (V.R.); drcretoiu@gmail.com (D.C.); oana.toader@yahoo.com (O.D.T.); 2Department of Obstetrics and Gynecology, Carol Davila University of Medicine and Pharmacy, 020021 Bucharest, Romania; nicolaebacalbasa@gmail.com (N.B.); anca.panaitescu@umfcd.ro (A.M.P.); ciobanu.ancamarina@gmail.com (A.M.C.); 3Department of Obstetrics and Gynecology, Filantropia Clinical Hospital, 011171 Bucharest, Romania

**Keywords:** fetal hydrops, mirror syndrome, heart failure

## Abstract

Mirror syndrome (MS) or Ballantyne’s syndrome is a rare maternal condition that can be life-threatening for both mother and fetus. The condition is characterized by maternal signs and symptoms similar to those seen in preeclampsia in the setting of fetal hydrops. Despite recent advances in the field of maternal-fetal medicine, the etiopathogenesis of MS remains elusive. For patients and doctors, the COVID-19 pandemic has become an extra hurdle to overcome. The following case illustrates how patients’ non-compliance associated with mirror syndrome and SARS-CoV-2 infection led to the tragic end of a 19-year-old patient. Therefore, knowledge of the signs and symptoms of mirror syndrome should always be part of the armamentarium of every obstetrician.

## 1. Introduction

Ballantynes syndrome, also known as mirror syndrome, was first described in 1892 and is characterized by maternal edema/hydrops following fetal hydrops [1]. This maternal pathology, which reflects fetal condition, has been given several names over the years, such as acute second-trimester gestosis pseudotoxemia, pregnancy toxemia, maternal hydrops syndrome, early-onset preeclampsia, triple edema, or mirror syndrome [2]. Likely an under-diagnosed condition, it can become life-threatening. Its substantial similarity with preeclampsia can mislead clinicians and delay treatment.

As COVID-19 continues, it greatly impacts healthcare services and especially prenatal care. Older age and underlying conditions substantially increase the fatality rate, but pregnant patients are no exception, being at increased risk of severe complications. For many patients, restrictive measures, social life decline, the concern of getting infected, changes in healthcare services, and fear of substandard antenatal care have caused excruciating anxiety. Psychological support and proper prenatal care are essential for a positive pregnancy outcome [3].

Our article aims at describing the rare association of pregnancy, mirror syndrome, and COVID-19 that ended in a tragic outcome.

## 2. Case Presentation

We report a case of a 19-year-old patient, 26 weeks primigravida, Rhesus positive, and a body mass index (BMI) of 31 who self-referred to our department with signs and symptoms of preterm labour. We noticed that she did not attend several appointments from her antenatal notes, including her second-trimester anomaly scan. Nevertheless, she underwent amniocentesis in a different unit at 16 weeks of gestation for fetal hydrops, according to local guidelines [4]. Molecular karyotyping (array Comparative Genome Hybridization—CGH) was carried out, using SurePrint G3 CGH ISCA v2 SNP-Microarray, 4 × 180 K (Agilent technologies), and showed a normal result. Following this procedure, she missed all of her other appointments with midwives and doctors alike. She presented anemia, hypoproteinemia, increased uric acid levels, and abnormal liver function, but normal blood pressure, and no proteinuria on current admission. Her COVID-19 RT PCR test was negative. She did report excessive weight gain (10 kg) over the last two weeks before the presentation. Her clinical examination revealed mild pitting pedal edema, no headache, no blurred visions, and right upper quadrant pain. Bilateral deep patellar reflexes were normal, and no ankle clonus was shown. She was contracting at every 15 min, but her cervix was long and closed. An ultrasound scan performed in our unit at 26 weeks demonstrated a live fetus with generalized hydrops, severe ascites, skin edema, severe polyhydramnios (amniotic fluid index of 36 cm), lack of fetal movement, and a very thick and large placenta (Figure 1, Figure 2, Figure 3 and Figure 4). Following comprehensive counselling regarding the fetal prognosis and potential maternal complications, the parents chose to carry on with the pregnancy. During the second day of admission, as the patient reported increasing shortness of breath and uterine discomfort, amniodrainage was performed and approximatively 1.5 L of clear amniotic fluid was removed with a subsequent, but short, improvement in maternal condition and fetal movements. Twenty-four hours later, she complains of occipital headaches, blurred vision, proteinuria, minimal urine output, and blood pressure of 178/110 mmHg. Considering her features of preeclampsia, she was started on intravenous magnesium sulphate (MgSO_4_), dexamethasone, and antihypertensive treatment followed by an emergency cesarean section. The patient delivered a stillborn female fetus weighing 1600 g and a large edematous placenta weighing 800 g. She received three units of fresh frozen plasma, three units of packed red cells, and albumin. Her condition deteriorated following a short improvement, accusing chest pain, increased dyspnea, diminished O_2_ saturation, dry cough, and worsening blood values. Urine output was less than 500 cc/day. Her hemoglobin levels, liver, and kidney function worsened in less than 24 h (Table 1). Clinical examination findings included cyanosis, increased jugular venous pressure, tachycardia, and bilateral pulmonary crackles due to pulmonary edema. Maternal echocardiography showed pericardial effusion, a 2 cm ventricular septal defect (VSD), a large ostium primum atrial defect, an elevated pulmonary artery diastolic pressure, an elevated right ventricular systolic pressure, and right atrial pressure, reflecting pulmonary hypertension. The ejection fraction was less than 30%. Upon a diagnosis of Eisenmenger syndrome, a decision was made to transfer the patient to a highly specialized Intensive Care Unit (ICU). Prior to the transfer, a COVID-19 real-time polymerase chain reaction test (RTPCR) was performed, which turned out to be positive. Nevertheless, she was transferred to a specialized COVID ICU under the care of a multidisciplinary team, where heart failure treatment was initiated. She was started on a high-flow nasal cannula and continuous renal replacement therapy (CRRT). Later on, she developed acute respiratory distress syndrome (ARDS), which required mechanical ventilation. Unfortunately, despite best efforts, she passed away at 14 days postpartum. A post mortem examination was not done as it is not customary to perform it in SARS-CoV-2 patients.

Macroscopic assessment of the fetus showed a normal morphological baby, excepting features consistent with hydrops fetalis. Fresh and old thrombi were found in medium and small vessels. Placental histology showed an increased placental mass, intervillous fibrin, and focal edema. Recent thrombosis was noted in the fetal umbilical vein as well.

## 3. Discussion

Mirror syndrome is uncommon and underreported, so the true incidence is difficult to ascertain. Despite recent advances in ultrasound and genetics, its pathology and etiology have yet to be determined. A variety of reasons leading to non-immune hydrops are described, such as chromosomal abnormalities, infection, fetal arrhythmias, and structural abnormalities (aneurysm of Galen’s vein, sacrococcygeal teratoma) [4]. There are still many cases where the etiology of fetal hydrops/mirror syndrome has not been clarified. Although no conclusive cause of Mirror syndrome has been found, the edematous placenta is alleged to play a provocative role in the appearance of maternal edema mediated through circulating angiogenic factor [5]. It is not easy to differentiate between mirror syndrome and preeclampsia. Clinical appearances of this illness are complex, including weight increase and edema, complemented by elevated blood pressure, with or without proteinuria, tachycardia, tachypnea, and oliguria. Laboratory values typically show signs of hemodilution with mild or moderate anemia, hypoalbuminemia, hyperuricemia, and higher human chorionic gonadotrophin (hCG) levels. Therefore, this disease can be simply mistaken for preeclampsia, making it difficult to diagnose and manage. This patient developed hydrops fetalis early in the pregnancy at 16 weeks of gestation. According to the literature reports, the gestational age for the onset of this syndrome ranges from 16 to 34 weeks [2]. Due to poor compliance, the patient missed all of her further appointments until 26 weeks, when she presented in our unit. This is a classic situation of severe fetal hydrops that can cause maternal complications like “mirror syndrome”. With standard prenatal care, it is reasonable to believe that mirror syndrome would have been identified at an earlier gestational age, although probably with the same outcome, as the patient decided to continue the pregnancy until an emergency cesarean section was required. In several reported cases, treatment and improvement of fetal hydrops have led to the resolution of maternal symptoms [6,7]. The decision to start MgSO_4_ implies delivery of the fetus. This was a sensible decision as the mother’s condition worsened over the night despite amniodrainage. In mirror syndrome, liver function and platelets are usually unaffected [8]. In our patients, liver enzymes have reached staggering values, making the differential diagnosis even more difficult. Mirror syndrome or preeclampsia is not a straightforward diagnosis. Many overlapping signs and symptoms might mislead clinicians. Hydrops fetalis was noticed at 16 weeks of gestation. At the time of admission, she presented various signs and symptoms, which could lead physicians toward one diagnosis or another. There was no proteinuria and the blood pressure was normal. Upon admission and amniodrainage, she did develop raised blood pressure and a clinical picture suggestive of preeclampsia. Therefore, MgSO_4_ protocol was initiated. Looking at the chronology of events starting at 16 weeks of gestation, admission time, caesarean section, and postpartum events, we concluded that this case started initially as a mirror syndrome. Generally, maternal symptoms tend to disappear shortly after successful management of fetal hydrops or removal of the placenta. This was not our case [2]. At the time of cesarean section, the patient was already in a life-threatening condition with liver and kidney insufficiency. Eisenmenger syndrome diagnosis came as an extra burden on an already critical patient. A low left ventricular ejection fraction (LVEF) is associated with a mortality of between 9 and 30% [9,10]. Neither the doctors nor the patient were aware of the large VSD. Chest X-ray, ECG, and echocardiogram were not performed on admission. Bearing in mind the diagnosis mentioned above and the continuous deterioration of the patient’s condition, a decision was made to transfer the patient to ICU in a different hospital, where a coat was available. Although her RT PCR COVID-19 test was negative at admission, a second test was performed prior to her transfer, which turned out to be positive. During her admission in our unit, she did not present any apparent signs of COVID-19 infection. The negative PCR test and her clinical picture suggestive for mirror syndrome/preeclampsia might have misled clinicians. Pregnancy and obesity represent risk factors for severe illness from COVID-19. Pregnant women have a higher chance of death, pneumonia, and ICU admission, as shown by Poon et al. [11,12]. When SARS-CoV-2 infection was diagnosed, our patient already met criteria for ICU transfer. As soon as she arrived in the ICU, she was started on heart failure treatment, CRRT, high flow canula, supportive treatment, and later, on mechanical ventilation. Her hepatic function worsened during hospitalization. COVID-19 predominantly affects the pulmonary tract causing mainly respiratory symptoms; nevertheless, involvement of other organs have been described, including the liver. In a study published by Yadav et al., liver biochemistry abnormalities were noted in as many as 76% of cases [13]. The patient’s liver function test was already affected on admission. Most likely elevated liver enzymes were the result of multiple conditions affecting the patient. In patients with Eisenmenger syndrome, hepatic alterations are the consequence of cyanosis and hypoxemia due to the intra cardiac right-to-left shunt, low cardiac output with inadequate liver perfusion, and liver congestion in right heart failure or chronically elevated central venous pressure [14]. While COVID-19 can affect liver function as well, we believe that, in our case, the patient’s progress towards hepatic insufficiency would have taken place even in the absence of SARS-CoV-2 infection, as her situation was already life-threatening. The highest risk lies in a patient with Eisenmenger syndrome during delivery and the early postpartum period due to large hemodynamic changes. Presumably, this is what lead to the sudden deterioration of her condition. The mortality rate for pregnant women with Eisenmenger syndrome is around 30–50%. For a patient diagnosed with additional conditions, these figures are even higher [15,16]. Our patient was diagnosed with mirror syndrome on admission. Later during hospitalization, her condition evolved to preeclampsia, hepatic insufficiency, and Eisenmenger syndrome. SARS-CoV-2 infection was rather an incidental finding, and it is difficult to establish how much COVID-19 contributed to the patient’s collapse, if it contributed at all. Considering the diagnosis mentioned above, we can conclude that this critically ill patient had an extremely guarded prognosis even in the absence of COVID-19 infection. There are a lot of unknown answers in terms of organ pathology and system insufficiency in this patient. Unfortunately, an autopsy, which could have brought up more information, was not performed as this is the protocol for COVID-19 patients. Previous articles have described the association between mirror syndrome and heart failure. While in the case described by Xu et al., the patient had unspecified heart surgery in childhood, in the case published by Li, there was no obvious cause of heart failure [5,17]. Though the etiology of mirror syndrome remains obscure, it is sensible to consider that mirror syndrome and its multiple effects such as pulmonary edema, anemia, hemodilution, and hypoalbuminemia can only worsen the outcome of patients with heart abnormalities.

## 4. Conclusions

For healthcare professionals involved in caring for pregnant women, is of utmost importance to understand complex interactions between pregnancy and different medical disorders. Further research is necessary to clarify the intricate relationship between fetal hydrops, placental edema, and mirror syndrome. Fetal hydrops and its possible complications should be part of the armamentarium of any physician involved in the care of pregnant patients. As our case illustrates, it can have fatal consequences not only for the fetus, but also for the mother.

The pandemic caused by SARS-CoV-2 is affecting people and healthcare structures worldwide. It is clear now that the cumbersome diagnosis of COVID-19 extends way beyond the illness of contracting the virus. To the best of our knowledge, this is the first case of mirror syndrome complicated by Eisenmenger syndrome and, to some extent, by COVID-19. As the disease continues, more data are becoming available regarding illness severity, cures, and treatment.

## Figures and Tables

**Figure 1 medicina-57-01031-f001:**
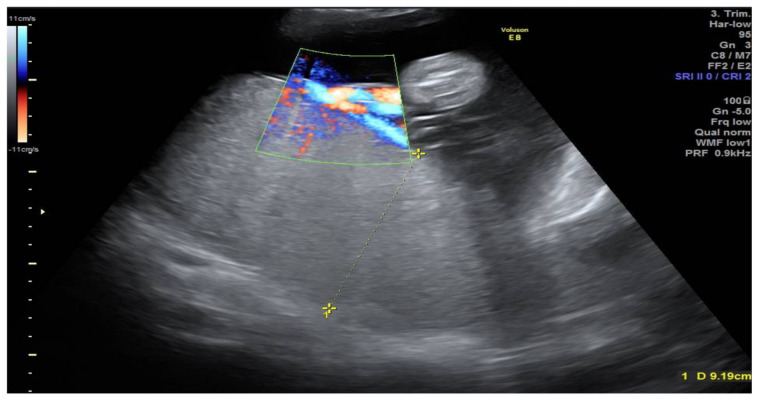
Ultrasound image of placenta—Placentomegaly.

**Figure 2 medicina-57-01031-f002:**
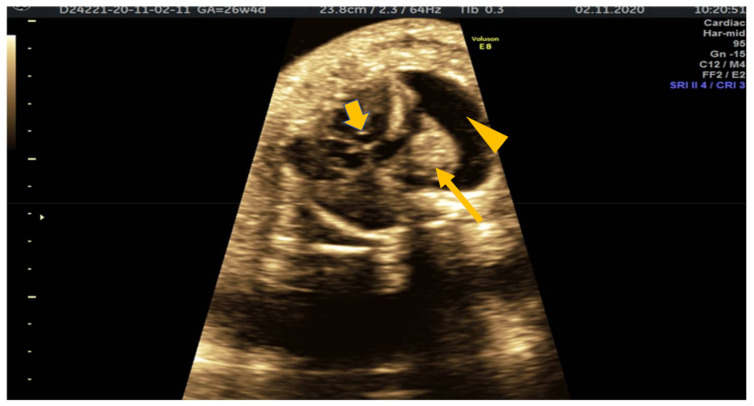
Transverse thoracic section showing fetal heart (short arrow), lung (long arrow) and pericardial effusion (arrowhead).

**Figure 3 medicina-57-01031-f003:**
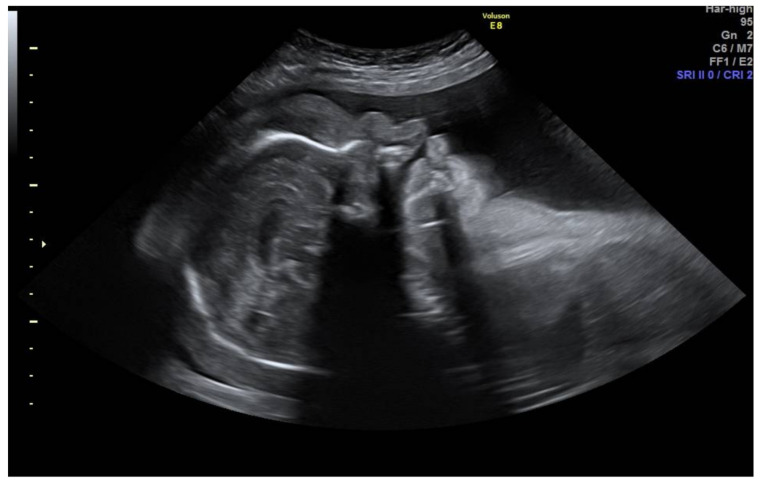
Profile section showing frontal bossing, massive scalp, and face edema.

**Figure 4 medicina-57-01031-f004:**
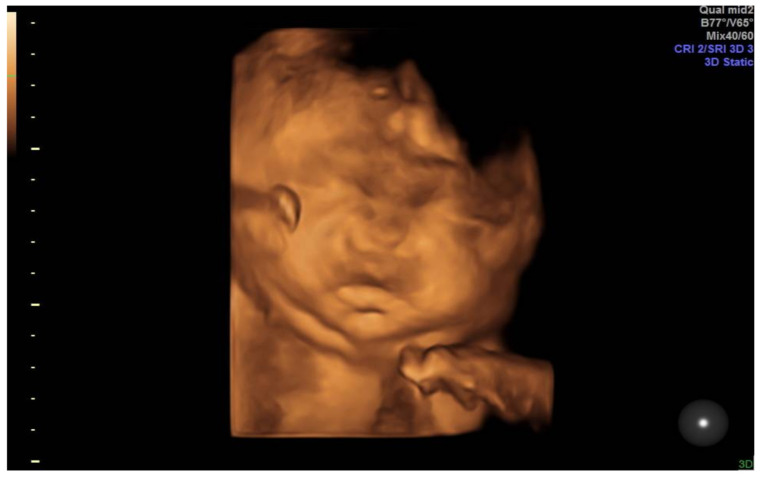
3D rendering with massive face edema.

**Table 1 medicina-57-01031-t001:** Blood results during hospitalization.

Blood Constant	Hb g/dL	Uric Acid mg/dL	ALT μ/L	AST μ/L	Creatinine mg/dL	WCC	CRP mg/dL
Pregnacy range	10.5–14.8	0–5.7	0–30	0–30	0.7–1.2	6–13,000	0.7–0.9
Admission D1	8.2	6.8	114	124	1.69	10,000	3
Surgery Day D2	7.7	8.8	759	1015	1.79	14,000	7
Post Surgery D3	9.2	8.4	701	1213	2.01	22,000	12

D1 = day 1, D2 = day 2, D3 = day 3, Hb = hemoglobin, AST = aspartate aminotransferase, ALT = alanine aminotransferase, WCC = white cell count, CRP = C-reactive protein.

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
