# Peer review of "Fatal Association of Mirror and Eisenmenger Syndrome during the COVID-19 Pandemic"

_medicina, 2021, doi:10.3390/medicina57101031_

Round 1
Reviewer 1 Report
The authors present a case of fetal hydrops in a pregnancy complicated by an undiagnosed ventricular septal defect. The patient's general condition deteriorated after admission, and an emergency cesarean section was performed. After surgery, an undiagnosed ventricular septal defect was found, resulting in Eisenmenger's syndrome, and COVID-19 infection was also found. Unfortunately, the patient passed away 14 days after the surgery.
There is a complex relationship between the following factors: fetal hydrops, undiagnosed congenital heart disease, systemic conditions like pre-eclampsia, and COVID-19 infection. Unfortunately, these relationships and Interpretations are not fully explained by the authors.
Case
- Line 46: The patient presented with symptoms of impending preterm labor, but there was no information on cervical length or uterine contractions.
- Line 57: Were there any physical findings or vital signs suggestive of heart failure at the time of the visit?
- Line 58: The investigation of the cause of fetal hydrops has not been explained. Were there any fetal infections (TORCH), hemolysis, morphological abnormalities, or fetal heart disease (arrhythmia, cardiomyopathy)?
- Line 105: Have you performed a pathological autopsy on a stillborn baby? Did you find any findings?
Figures
- Figure 2: I could not determine where the fetal pericardial effusion was. Would you please present heart, lung, pleural and pericardial effusion?
- Figure 3 and 4: What is the significance of presenting Figures 3 and 4 in this case report? For subcutaneous edema, I think Figure 2 is sufficient.
Discussion
- The authors said that it is not easy to differentiate between mirror syndrome and preeclampsia. However, the authors concluded that this case was mirror syndrome. The reason for the diagnosis of mirror syndrome rather than early-onset preeclampsia or pregnancy with heart failure is insufficient.
- Line 157: The authors suggested that the cause of elevated liver enzymes may be COVID-19 infection. Has elevated liver enzymes been reported to be seen even when the PCR test for COVID-19 is negative? What is the reason for your conclusion that elevated liver enzymes are not due to preeclampsia or undiagnosed heart failure by VSD?
- Is COVID19 strongly involved in the outcome of this case? For example, do the authors believe that COVID19 is involved in the deterioration of the general condition after hospitalization and/or the development of Eisenmenger syndrome after surgery? If so, please provide reasons for this belief.
Author Response
Thank you for reviewing this manuscript and thank you for sharing your thoughts in connection with the paper. We have made corrections and revised the text in line with your suggestions and we believe this has assisted to substantially improve the clarity of the text and the overall quality of the paper.
Enclosed, please find responses to your individual points and the revised manuscript.
Sincerely yours, Lucian Pop MRCOG

Reviewer 2 Report
The authors selected a case report which covers rare and new diseases with a multidisciplinary approach. The title is relevant and includes keywords related to the case. The design is suitable and context is adequately described. The text is clear and easy to read. The methods are thorough and the results reflect a challenging differential diagnosis in unusual settings. The weakness of this report is a missing postmortem examination which was also stressed out by authors. The use of graphs and figures are adequate. References are of recent publications. In summary, the article is very interesting and I recommend it for publication.
Author Response
Dear reviewer,
Thank you very much for all your comments and your recommendations.
The authors
Round 2
Reviewer 1 Report
The authors have responded appropriately to most of my comments. However, the authors need to describe more about the relationship between the pathologies; mirror syndrome, heart failure, and COVID-19.
I speculate that in this case, Miller's syndrome is involved in the worsening of heart failure and the development of Eisenmenger's syndrome. I assume that the heart failure was not detected because the chest examination findings and subsequent chest x-ray, ECG, and echocardiogram were not performed adequately on admission. There are some case reports discussing mirror syndrome and heart failure.
PMID: 30640274, 26629125
I also think the title is misleading; while I agree that COVID19 put an extra burden on her, it is unclear whether COVID19 was involved in the pathogenesis of this case. Therefore, I suggest that the author revise the title appropriately.
Author Response
Dear reviewer,
Once again, thank you very much for your suggestions
Point by point answer: 1. While Mirror syndrome can have serious consequences in a patient without any other obvious condition, in a patient with heart abnormalities prognosis is worse. This idea is stressed out on pg 6 line 42 and on pg 7 line 12 ( second revision)
2. Heart abnormalities were not detected on admissions, this is written more clearly on pg 6 line 20.
3. We concur with the idea that is unclear how much COVID 19 contributed to the fatal outcome. Regrettably, COVID 19 precluded an autopsy which could have given us more answers. The title of the paper was changed appropriately to : Fatal association of mirror and Eisenmerger syndrome during the COVID-19 pandemic.
We hope that we managed to respond to the concerns in a satisfactory manner
